# Melatonin Reduces Oxidative Stress in the Right Ventricle of Newborn Sheep Gestated under Chronic Hypoxia

**DOI:** 10.3390/antiox10111658

**Published:** 2021-10-22

**Authors:** Alejandro Gonzaléz-Candia, Pamela V. Arias, Simón A. Aguilar, Esteban G. Figueroa, Roberto V. Reyes, Germán Ebensperger, Aníbal J. Llanos, Emilio A. Herrera

**Affiliations:** 1Laboratory of Vascular Function and Reactivity, Pathophysiology Program, ICBM, Faculty of Medicine, Universidad de Chile, Av. Salvador 486, Santiago 7500922, Chile; alejjobq@gmail.com (A.G.-C.); pamelaariasalessandrini@gmail.com (P.V.A.); s.aguilar.osorio96@gmail.com (S.A.A.); s.teban.fb@gmail.com (E.G.F.); 2Institute of Health Sciences, University of O’Higgins, Libertador Bernardo O’Higgins 611, Rancagua 2820000, Chile; 3Pathophysiology Program, ICBM, Faculty of Medicine, Universidad de Chile, Av. Salvador 486, Santiago 7500922, Chile; vreyes@uchile.cl (R.V.R.); gebensperger@uchile.cl (G.E.); allanos@med.uchile.cl (A.J.L.); 4International Center for Andean Studies (INCAS), Universidad de Chile, Baquedano s/n, Putre 1070000, Chile

**Keywords:** antioxidant enzymes, pro-oxidant sources, right ventricular remodeling, pulmonary arterial hypertension of the neonate, melatonin receptors

## Abstract

Pulmonary arterial hypertension of newborns (PAHN) constitutes a critical condition involving both severe cardiac remodeling and right ventricle dysfunction. One main cause of this condition is perinatal hypoxia and oxidative stress. Thus, it is a public health concern for populations living above 2500 m and in cases of intrauterine chronic hypoxia in lowlands. Still, pulmonary and cardiac impairments in PAHN lack effective treatments. Previously we have shown the beneficial effects of neonatal melatonin treatment on pulmonary circulation. However, the cardiac effects of this treatment are unknown. In this study, we assessed whether melatonin improves cardiac function and modulates right ventricle (RV) oxidative stress. Ten lambs were gestated, born, and raised at 3600 m. Lambs were divided in two groups. One received daily vehicle as control, and another received daily melatonin (1 mg·kg^−1^·d^−1^) for 21 days. Daily cardiovascular measurements were recorded and, at 29 days old, cardiac tissue was collected. Melatonin decreased pulmonary arterial pressure at the end of the experimental period. In addition, melatonin enhanced manganese superoxide dismutase and catalase (CAT) expression, while increasing CAT activity in RV. This was associated with a decrease in superoxide anion generation at the mitochondria and NADPH oxidases in RV. Finally, these effects were associated with a marked decrease of oxidative stress markers in RV. These findings support the cardioprotective effects of an oral administration of melatonin in newborns that suffer from developmental chronic hypoxia.

## 1. Introduction

Hypoxia is one of the most severe challenges during perinatal development [1]. The response to hypoxia involves complex intracellular networks that may be derived from cardiovascular diseases, such as pulmonary arterial hypertension, affecting fetuses and newborns [2]. Chronic hypoxia during gestation results in the pulmonary vasculature failure to dilate adequately and decrease the arterial wall thickness after birth, leading to pulmonary arterial hypertension, hypoxemia, and the remodeling of the right ventricle (RV) and, ultimately, resulting in the syndrome of pulmonary arterial hypertension in the newborn (PAHN) [3]. This is a relevant health problem, considering there are an estimated 170 million permanent residents at altitudes >2500 m, mainly on the Tibetan, the Andean, and the East African plateaus [4,5]. In addition, it is estimated that 3–4% of lowland pregnancies are exposed to chronic hypoxia due to materno-placental or fetal complications which may also result in PAHN [6].

In PAHN, the RV undergoes prolonged pressure overload conditions. Firstly, and as an adaptive mechanism to maintain the pulmonary circulation, the RV induces a hypertrophic remodeling (concentric hypertrophy), characterized by an increased cardiomyocyte (CM) size and preserved RV function [3]. This myocardial remodeling is associated with an increased intrinsic contractility and capillary angiogenesis in the myocardium, providing sufficient oxygen and nutrients for ventricular function [7].

However, if pressure overload remains (as seen in PAHN) the adaptive response gradually shifts to a maladaptive remodeling towards RV failure [8]. The cellular responses that drive RV hypertrophy in PAHN depend not only on the severity of the pulmonary vascular disease, but also on the interplay between several mechanisms such as changes in myocyte gene expression [9], transition to glycolytic metabolism [10], neurohormonal activation with upregulation of the renin–angiotensin system [11], adrenergic overstimulation, and increased expression of several counter-regulating peptides (e.g., natriuretic peptides). All of the above influences the function of cardiac myocytes. For instance, contractile dysfunction is associated with an α to β-isotype switch of the major thick filament protein myosin heavy chain (MHC) (α-MHC/β-MHC switch) in cardiomyocytes, decreasing the cardiac systolic contractile response [10,12,13].

Oxidative stress induced by chronic hypoxia plays a crucial role in RV remodeling [14]. In addition, it has been shown that the RV is more vulnerable to oxidative stress than the left ventricle (LV) [15], probably due to the limited capacity of the RV to upregulate antioxidant enzymes such as superoxide dismutase isoforms [16]. The latter decreases the ability of the RV to neutralize the reactive oxygen species (ROS) generated by chronic hypoxia. On the other hand, ROS can be generated in the RV by various mechanisms. One important mechanism is the oxidative phosphorylation in the mitochondria as a byproduct of normal cellular aerobic metabolism [17,18]. During hypoxia, the mitochondrial electron uncoupling at complexes I and III of the electron transport chain enhances the superoxide anion (O_2_^●−^) generation [17]. Other important cardiac ROS sources are the xanthine oxidase (XO), NAD(P)H oxidases, cytochrome P450, autooxidation of catecholamines, and uncoupling of NO synthase (NOS) [18,19,20]. NO contains an unpaired electron and, under certain conditions, can react with O_2_^●−^ to form peroxynitrite, a potent oxidant, classified as a reactive nitrogen species (RNS). Excessive production of ROS and RNS induces contractile dysfunction by suppressing enzymes involved in excitation–contraction coupling and through polynitrosylation of the ryanodine receptor, increasing the bioavailability of intracellular calcium [21]. Additionally, ROS and RNS favor cardiac remodeling through antioxidant enzymes inactivation and the induction of apoptosis and inflammation as the main mechanisms of cell damage [22].

Several mechanisms counterbalance the production of ROS, including enzymatic and nonenzymatic cellular pathways. Among the best-characterized enzymatic pathways are the superoxide dismutases (SODs), which facilitate the formation of H_2_O_2_ from O_2_^●−^, and catalase (CAT) and glutathione peroxidase (GPx), which converts H_2_O_2_ to water [18]. Moreover, the enzymatic and nonenzymatic antioxidant systems are potentially depleted by exposure to chronic hypoxia such as glutathione peroxidase, superoxide dismutase activities, and the total antioxidant capacity [23].

Melatonin is an indoleamine secreted by the pineal gland, which has been shown to add protection against ROS. This protection is based on at least three known properties: a direct ROS scavenger [24], the stimulation of antioxidant enzymes [25], and a negative modulation of ROS generation in pro-oxidant sources [26]. Melatonin typically interacts with membrane melatonin receptors (MT1 and MT2) and nuclear receptors of the retinoic acid receptor-related orphan receptor (ROR) family [27]. These receptors mediate a myriad of effects, including changes in intracellular cyclic nucleotides (cAMP, cGMP) and calcium signaling, activation of specific protein kinase C subtypes, modulation of activity and intracellular localization of steroid hormone receptors, and G protein coupled signaling [28]. Melatonin exerts its physiological action in cerebral areas (i.e., the hippocampus, cerebellum, and prefrontal cortex), by acting through membrane MT1 or MT2 receptors. In addition, in the suprachiasmatic nucleus (SCN) and hippocampus, these receptors are implicated in the regulation of sleep and circadian rhythms. [28,29]. Moreover, previous evidence suggests versatile melatonin roles in the cardiovascular system, particularly cardioprotection against ischemic heart injury and cardiomyopathy [30]. Consistently, blunted melatonin secretion deteriorates hypertension and cardiac hypertrophy, while melatonin administration effectively protects the heart from pathological hypertrophy [31]. For instance, melatonin induces the relaxation of pulmonary arteries and veins [32,33], which may favor RV function. In general, MT2 receptors expressed in SMC seem to mediate vasodilation, while MT1 induces vasoconstriction [34]. Thus, we hypothesized that neonatal melatonin administration could regulate the redox balance in the right ventricle of chronically hypoxic and pulmonary hypertensive neonatal lambs. Furthermore, we examined whether melatonin modulates the expression and activity of antioxidant enzymes as well as pro-oxidant sources in RV.

The neonatal lamb model has been used as a model for lung development and pulmonary function for several reasons reviewed elsewhere [35]. One important characteristic is that the lung development in sheep is similar to humans with respect to timing and birth maturation. In addition, the fetal to neonatal cardiopulmonary transition represents a good paradigm for perinatal pulmonary transition in newborn humans [36].

## 2. Materials and Methods

### 2.1. Animals and Treatment

All animal care, maintenance, and experimental procedures were approved by the Bioethics Committee of the Faculty of Medicine, University of Chile (CBA #0761 FMUCH) and were carried out according to international standards following the Guide for the Care and Use of Laboratory Animals published by the US National Institutes of Health (NIH Publication No. 85–23, revised 1996).

Ten newborn sheep (*Ovis aries*) were gestated, born, and studied at a high altitude (Putre, 3600 m), randomly divided into two groups. The control group (C) received vehicle (*n* = 5, 1 female and 4 males, 1.4% ethanol 0.5 mL·Kg^−1^, orally) and the treated group received melatonin (M) (*n* = 5, 3 females and 2 males, melatonin 1 mL·Kg^−1^ in 1.4% ethanol 0.5 mL·Kg^−1^, orally) daily for 21 days. The dose of 1 mL·Kg^−1^ of melatonin is based on previous pharmacokinetic data from PAHN sheep neonates without generating adverse effects in the experimental model [37]. To maintain the physiological endogenous rhythm of melatonin, the daily doses were given at dusk (at 20:00). Five days after treatment termination (29 days old), lambs underwent euthanasia with an overdose of sodium thiopentone (100 mL·Kg^−1^, slow I.V. infusion) and RV tissue was obtained for in vitro studies.

### 2.2. Cardiopulmonary Measurements

Lambs were cathetherized at three days old under general anesthesia (ketamine 10 mL·Kg^−1^–xylazine 0.04 mL·Kg^−1^, IM) and aseptic procedures [38,39]. From day 3 onwards, we performed daily determination of the pulmonary arterial pressure (PAP) and heart rate (HR) (between 9:00 and 11:00 AM for 30 min) with an acquisition system (PowerLab/8SP System and Chart v4.1.2 Software System, ADInstruments, Sidney, Australia) connected to a computer [37,39]. The cardiac output (CO) was determined in triplicate with the Swan–Ganz catheter connected to a cardiac output computer (COM-2 model, Baxter, Irvine, CA, USA). In addition, pulmonary vascular resistance (PVR) was calculated as described previously [38]. All in vivo measurements were performed on conscious and unanesthetized animals resting comfortably in a canvas sling.

### 2.3. Cardiac Histomorphometry

Immediately after extraction, heart tissue was washed in warm PBS and cut in 1 cm thickness slices. Then, the fresh slices were photographed and the heart walls and chambers area were measured as described elsewhere [40]. Briefly, the analyses consisted in determining luminal ventricular areas and thicknesses of both the ventricular free walls and septum [41]. Thereafter, heart tissue was immersed fixed with 4% PFA for 24 h at 4 °C and kept in PBS. Fixed samples were embedded in paraffin and cut into 4 μm slides. Hematoxylin–Eosin staining was performed in images captured at 10× with a digital camera coupled to a microscope (Olympus BX-41). The microphotographs analysis was performed with the software Image Pro-Plus 6.2 (Media Cybernetics Inc., Rockville, MD, USA). All analyses were performed by two independent observers who were blind to the treatment groups.

### 2.4. Antioxidant Enzyme Expression and Activity in the Right Ventricle

Protein expression of MnSOD, CAT, GPx1 and β-actin was determined in RV lysates by immunoblot using specific antibodies (anti-Mn-SOD, Millipore, 06-984, 1:1000; anti-CAT, Abcam Laboratories, ab1877, 1:10,000; anti-GPx1, Abcam Laboratories, ab22604, 1:1000; and anti-β-actin, AC-15, Thermo Fisher scientific, 1:5000), as described elsewhere [34]. The signals obtained by immunoblot were scanned and quantified by densitometric analysis (Odyssey Imaging System, Li-Cor Biosciences, Lincoln, NE, USA).

The antioxidant enzymes activities in RV tissue homogenate were measured using the Superoxide Dismutase (SOD) Activity Assay Kit (K335-100, Biovision, quantified at 450 nm), the OxiSelect Catalase Activity Assay Kit (STA-341, Cell Biolabs Inc., quantified at 520 nm), and the Glutathione Peroxidase Assay Kit (703102, Cayman Chemical Company, Ann Arbor, MI, USA, quantified at 350 nm), according to the manufacturers’ guidelines. Total protein concentration was used for normalization purposes [26].

### 2.5. Quantification of Pro-Oxidant Sources in the Right Ventricle

Protein levels of voltage-dependent anion-selective channel (VDAC), p47-phox, Xanthine Oxidase (XO), and α-tubulin was determined in VD lysates by immunoblot with specific antibodies (anti-VDAC, Millipore, AB10527, 1:1000; anti-p47-phox, Sigma Aldrich, SAB45028110, 1:1000; anti-Xanthine oxidase, Abcam, ab125133, 1:1000 and anti-α-tubulin, cell signaling, # 2148, 1:2000, respectively), as described previously [34]. The signals obtained on immunoblot determinations were scanned and quantified by densitometric analysis with a chemiluminescence detection device (Odyssey Imaging System, Li-Cor Biosciences, Lincoln, NE, USA).

Mitochondrial O_2_^●−^ generation was quantified in the RV. Briefly, RV tissue was homogenized in five volumes of buffer containing sucrose 70 mmol/L, mannitol 210 mmol/L, HEPES 5 mmol/L, EGTA 1 mmol/L, and BSA 0.5% (*p/v*), pH 7.2, at 4 °C. The homogenized tissue was centrifuged at 1000× *g* for 10 min at 4 °C, and the supernatant was collected and centrifuged at 12,000× *g* for 10 min. Then, the sediment was washed and centrifuged at 10,000× *g* for 10 min, and resuspended in a buffer containing sucrose 70 mmol/L, mannitol 210 mmol/L, and HEPES 5 mmol/L, pH 7.2, at 4 °C. This resuspension was immediately used to assess mitochondrial O_2_^●−^ production by measuring the oxidation of DHE 10 μmol/L (470Ex/590Em) during 10 min at 37 °C and expressed as relative fluorescence units (RFU) [26].

Nicotinamide adenine dinucleotide phosphate (NADPH) oxidase activity was determined based on the rate of consumption of NADPH monitored at 340 nm at 37 °C. The enzymatic reaction was initiated by adding 0.1 mmol/L NADPH and the activity was expressed in μmoles of oxidized NADPH/mg protein/min. Only slight oxidation of NADPH was detected in the presence of 100 μmol/L apocynin (data not shown), an inhibitor of NADPH oxidase [26].

Finally, the quantification of XO activity in the RV tissue was performed using a commercial XO assay kit (XO assay kit, ab102522, Abcam), following the manufacturer’s instructions. Briefly, the RV tissue was homogenized mechanically with four volumes of assay buffer (assay buffer^®^). Subsequently, the sample was centrifuged at 16,000× *g* for 10 min the supernatant was isolated. Thereafter, the probe (OxiRED™) was added to the supernatant and incubated at 37 °C for 1 h. Finally, the product of the reaction associated with a specific probe was quantified at an absorbance of 570 nm [26,42].

### 2.6. Oxidative Stress Markers

The oxidative stress level was determined by the 8-isoprostanes (8-iso-PGF2a) concentration in RV homogenates with a specific enzyme immunoassay kit following the manufacturer recommendations (Cayman Chemical, Ann Arbor, MI, USA). The ELISA kit’s detection limit was 3 pg/mL, and the estimated variability of the method was 4.2% for the inter-assay and 8.1% for the intra-assay. The product of the reaction was quantified at an absorbance of 410 nm [43].

### 2.7. Right Ventricle Immunohistochemistry

Oxidative stress marker 3-nitrotyrosine (3-NT) and melatonin receptor (MT1 and MT2) was assessed in-wall and endocardial vessels from the RV by the anti-nitrotyrosine monoclonal antibody (Clone 1A6, Millipore, Merck), MT1 (sc390328, Santa Cruz Biotechnology), and MT2 (sc398788, Santa Cruz Biotechnology). Briefly, the tissue sections were exposed to retrieval buffer 1X for antigen retrieval (Target Retrieval Solution, Dako) at 120 °C for 25 min. The primary antibodies anti-3-NT, anti- MT1 and anti-MT2 were incubated in bovine serum albumin 1% (1:100) for 3 h. Then the slides were incubated with an anti-mouse polymer (EnVision System-HRP, Dako) for 1 h. Finally, the immunoreaction was revealed with diaminobenzidine, and the nuclear stain was performed with Harris hematoxylin. All slides were digitally acquired at 40× (Olympus BX-41) and analyzed as specific brown reddish-pixel count per area relative to the positive control. For mark quantification, the intima and media-adventitia layers were selected and measured using Adobe Photoshop (CS5 extended version 12.0, San Jose, CA, USA). The mark intensity (pixels) was then divided by the area of each arterial layer (pixels/µm^2^) [44].

### 2.8. Statistical Analyses

All data were expressed as means ± SEM. In vivo measurements were expressed as daily averages. Cardiopulmonary data were analyzed by two-way ANOVA and the post-hoc test of Tukey. All other results were compared statistically by a Mann–Whitney t-test unless otherwise stated. Significant differences were accepted when *p* ≤ 0.05 (Prism 8.0; GraphPad Software).

## 3. Results

### 3.1. Cardiopulmonary Variables

Both experimental groups showed similar PAP, PVR, CO, and HR values before starting the treatment (Figure 1A–D). PAP decreased below 25 mmHg in the melatonin treated group and was less than in the control group (Figure 1A). Both, CO and HR showed a similar decrease with age in both groups whilst PVR remained unchanged (Figure 1B–D).

### 3.2. Cardiac Biometry

The macroscopic cardiac analysis (Figure 2) did not show significant differences between groups for the wall thicknesses for the right and left ventricles (Figure 2A,B) and the interventricular septum (Figure 2C). In addition, luminal areas for the right (Figure 2D) and the left (Figure 2D) ventricles were similar between groups.

### 3.3. Antioxidant Enzymes Expression and Activity in RV

Melatonin treatment increased the right ventricle MnSOD2 and CAT protein expression in M relative to C (Figure 3A,B). In contrast, similar levels were observed in GPx1 protein expression in both groups (Figure 3C).

Furthermore, similar SOD and GPX enzymatic activities were observed in M and C groups (Figure 3D,F) in RV. However, melatonin induced the enzymatic activity of catalase compared to the control group (Figure 3E).

### 3.4. ROS Sources in RV

Protein expression of VDAC (mitochondrial population) and XO were similar in RV for both groups (Figure 4A,C). In contrast, melatonin treatment decreased p47 phox subunit expression in RV (Figure 4B).

Also, melatonin markedly decreased the generation of O_2_^●−^ from mitochondria and NADPH oxidase (Figure 4D,E) in RV. In contrast, generation of O_2_^●−^ by XO did not show significant differences between groups (Figure 4F).

### 3.5. Oxidative Stress in RV

Melatonin treatment decreased oxidative stress in RV (Figure 5), measured by 8-isoprostanes in tissue homogenate (Figure 5A), 3-NT immunostaining in RV wall (Figure 5B) and 3-NT immunostaining in endocardial vessels (Figure 5C) relative to the control group.

### 3.6. Immunolocalization of Melatonin Receptors in the RV

The immunoreactivity for MT1 and MT2 receptor in the RV wall was similar between groups (Figure 6A,C). Similarly, MT1 expression in endocardial vessels was comparable among groups (Figure 6B). However, the melatonin treated group had a lower MT2 receptor immunoreactivity in endothelium compared to the control group, without changes in the vascular media-adventitia layers (Figure 6D).

## 4. Discussion

Our study describes the effects of postnatal melatonin administration on the redox balance of the right ventricle in an ovine model of PAHN. As the main cardiac effects are due to RV afterload and are associated with RV remodeling [11,12], we assessed changes in the right side of the heart. The data show that melatonin decreases right ventricle oxidative stress, mainly due to decreased mitochondrial and NADPH oxidase ROS generation and an increased catalase antioxidant activity. In addition, we demonstrate that melatonin receptors 1 and 2 are present in the right ventricle and that melatonin treatment modulates MT-2 expression in neonates. While postnatal treatment with melatonin protects the neonatal cardiac function and redox balance against the adverse effects of hypobaric pregnancies, it did not affect cardiac structure. Therefore, our findings support the hypothesis tested that neonatal melatonin administration regulates the redox balance in RV, protecting against cardiac dysfunction programmed by developmental hypoxia.

Previously, we have shown that pregnant ewes and fetal sheep at 3600 m are hypoxemic relative to lowland animals [45]. This condition induces a marked endothelial dysfunction and vascular remodeling in the neonatal lamb which results in PAHN [35,38,39,44]. Furthermore, we have shown that melatonin treatment for one week improves cardiopulmonary function and structure [39,44]. In addition, the extension of the treatment for three weeks at high-altitude decreased the vasoconstrictor function and increased the vasodilator function in small pulmonary arteries [37]. The latter was associated with a decreased pulmonary oxidative stress by inducing antioxidant enzymes and diminishing pro-oxidant sources [37]. However, until now, the effects of melatonin in the heart of neonates gestated, born, and raised under chronic hypoxia was unknown.

At the end of the in vivo recording (postnatal day 28), the group treated with melatonin decreased the PAP below 25 mmHg. This finding is relevant because pulmonary hypertension is diagnosed when PAP rises over 25 mmHg. In addition, neonates gestated and born in highlands have a delayed and incomplete pulmonary transition and progressive PAP decline is less (relative to healthy normoxic newborns) [46]. Therefore, the given treatment is clinically relevant.

Previous studies have shown that an oral administration of melatonin achieves both a high bioavailability and distribution volume, pharmacokinetics that predict high compartmental distribution and concentration in the lung and heart [37]. The possible mechanisms by which melatonin induces the fall of PAP at the end of treatment involve a decline of the vasoconstrictor tone, with a RV afterload and oxidative stress reduction occurring during the treatment. These mechanisms are supported by an increased pulmonary endothelium-dependent vasodilation and muscle-dependent vasodilation induced by the melatonin [37,39]. One likely explanation for this functional improvement is that melatonin diminishes oxidative tone in the cardiopulmonary circulation, improving the eNOS function and increasing NO bioavailability [37,39]. Besides, a decreased oxidative tone improves soluble guanylate cyclase (sGC) functionality by activating cGMP-dependent vasodilation [47].

During chronic hypoxia, the RV undergoes prolonged pressure overload. Initially, and as an adaptive response to preserve pulmonary blood flow, the RV endures hypertrophic remodeling, characterized by an increase in cardiomyocyte (CM) size, increased intrinsic contractility, and capillary angiogenesis providing sufficient oxygen and nutrients to the enlarged myocardium [48,49]. However, we did not find significant differences in cardiac output, heart rate, and structural (luminal areas and wall thicknesses) between the analyzed groups. We only found a progressive decrease in cardiac output and heart rate, evidenced when the initial and final treatment days were compared. This action is normal and is due to the disappearance of the umbilical–placental bed and the fetal vascular shunts [50,51]. Hypoxia is a main and strong insult that determines hypoxic pulmonary vasoconstriction (HPV) and RV increased afterload [52]. Therefore, this condition may be a possible explanation for the lack of response to melatonin in terms of cardiac structure. In our experimental design, we decided to evaluate the melatonin effect without oxygen supplementation. However, in the clinical setting, hypoxia was treated and reverted by oxygen supplementation [53], which presumably reverts HPV, decreases remodeling, and improves cardiac function simultaneously with melatonin’s beneficial effects.

ROS represents a set of reactive species derived from oxygen with one or more unpaired electrons, making them very unstable and highly reactive [54]. Therefore, ROS can bind and oxidize different cellular compounds (e.g., lipid, DNA, protein, and cellular membrane, among others), thus modifying their structure and function. One of the most novel findings in our study is the regulatory mechanisms exerted by melatonin on the antioxidant enzyme and main ROS sources at the RV tissue. Melatonin increased enzymatic expression of MnSOD and CAT. This has been previously described in neonatal pulmonary circulation [26,39], brain [55], and cultured cardiomyocytes [56]. Therefore, this study describes a further antioxidant regulation exerted by melatonin in neonatal hearts. Treatment with melatonin increased catalase activity, a key enzyme to decrease the availability of H_2_O_2_ produced by NADPH oxidase and XO enzymes induced by hypoxia. Catalase is an oxidoreductase that contains a heme group modified by oxidative stress, melatonin, and endogenous scavenging activity [57]. Melatonin may as well improve the cofactors of the catalase and increase its activity in VD. However, the total activity of SOD and GPx results from the sum of different isoforms of each enzyme that can be modified in different ways with respect to their function, expression, and structure by either melatonin or oxidative stress. Particularly, the dichotomy between SOD expression and activity can be explained from an experimental point of view since the blot (expression) involves only the mitochondrial isoform, and the activity kit involves the three isoforms. In this sense, it can be inferred that the expression of all three isoforms is differentially regulated. The cytoplasmic SOD1 isoform is practically constitutive according to Minc et al. [58], while the mitochondrial SOD2 and extracellular SOD3 isoforms are regulated by different stimuli, in particular the redox balance (i.e., NF-kB, AP-1, and AP-2) [59] and inflammation (particularly SOD) [60]. On the other hand, melatonin via a receptor (MTI or MT2) can regulate the activation of transcriptional factors such as AP-1 or NFkB [61]. Therefore, melatonin acts on SOD functional regulation. However, further studies need to be implemented to assess the expression of SOD and GPx isoforms in neonatal hearts after melatonin treatment.

Chronic hypoxia in RV induces ROS generation at the electron transporter chain, NADPH oxidase, and XO [62]. Melatonin treatment decreases the generation of O_2_^●−^ by the mitochondria without changes in the mitochondrial population determined by VDAC protein expression. Chronic hypoxia determines an incomplete reduction process on the final oxygen acceptor at the mitochondrial electron transport chain [63]. In this pathway, ubisemiquinone (UQ), flavin mononucleotide (FMN), and eight Fe-S (iron-sulfur) clusters seem to be responsible for ROS generation, both at complex I and III levels [64]. In addition, mitochondria are themselves susceptible to be damaged by these radicals. Therefore, we speculate that melatonin diffuses towards the mitochondria and acts scavenging free radicals generated by the decoupling of the respiratory chain in the mitochondrial crest [26,65]. Likewise, studies support that melatonin may induce, at the mitochondrial level, the increase of GSH, enhancing the mitochondrial antioxidant capacity [66]. Interestingly, melatonin strengthened cardiac endothelial cells by improving mitochondrial fusion [67] and blunting mitochondrial fission [68], eventually preventing cardiomyocyte apoptosis [69].

Another pro-oxidant source analyzed in this study was the NADPH oxidase (NOX), as the isoforms NOX1, NOX2, NOX4, and NOX5 are expressed in the cardiovascular system [70]. NOX is present in the plasma membranes of endothelial cells, cardiac muscle and fibroblasts, where it releases O_2_^●−^ that serves as a precursor for additional ROS including hydrogen peroxide, hydroxyl radical, peroxynitrite, and other oxidants [64]. Our results show a decrease in the O_2_^●−^ generation from NOX, associated with a decreased expression of p47 phox. The NOX activity assay used apocynin as NOX inhibitor to create a negative control. Whilst apocynin has been widely used as a NOX inhibitor in neurons, blood vessels, and hearts (among other tissues) [26,71,72], it has been also questioned because of its antioxidant activity per se [73]. We cannot completely rule out if apocynin acts as an antioxidant in addition to its action as NOX inhibitor in our assay. However, the similar 30–40% decrease of both NOX activity and expression induced by melatonin in the right ventricle suggests that our functional measurements are correct. Melatonin may impair NADPH oxidase assembly by inhibiting the phosphorylation of the p47phox subunit, avoiding p47phox and p67phox translocation to the membrane. [74]. Lastly, xanthine oxidase is the enzyme that catalyzes the conversion of hypoxanthine to xanthine and xanthine to uric acid, with simultaneous O_2_^●−^ generation [75]. Melatonin administration does not seem to modify XO levels or activity in the RV of our hypoxic neonates. However, an antenatal treatment with melatonin does decrease XO levels and activity in neonates [26]. The potential mechanisms by which melatonin regulates XO remains unknown [26].

Finally, this study quantified the immunoreactivity to melatonin receptors in cardiac muscle and endocardial vessels (intramyocardial vessels). Although there are no differences in the expression of melatonin receptors in cardiac muscles, the endothelial immunoreactivity in endocardial vessels showed a differential expression of MT-2 receptors. A previous study showed that the oral administration of melatonin to neonates may decrease melatonin receptor expression in the kidney but not in the heart [76]. To the best of our knowledge, this is the first report showing that melatonin may regulate its receptors in neonatal endocardial arteries. This finding is relevant because melatonin may be regulating the endothelial cells activation and therefore modifying vasodilator molecules such as nitric oxide (NO) or prostaglandin I2 (PGI2) [77]. Our findings may explain previous studies that described the direct vasodilator effects of melatonin [33,78]. In addition, melatonin modulates the release of NO, reducing peroxynitrite formation and offer protection from nitrosative/oxidative damage [78], as well as favoring an increased vasodilation in pulmonary arteries via COX1-PGIs-IP pathway [79]. Studies undertaken in several animal models have indicated that melatonin has dual effects on the vasculature, with vasoconstriction being observed through MT1 receptor activation and vasodilatation through MT2 receptor [28,80,81]. However, the role of melatonin in the right ventricular function has not been fully clarified.

In terms of oxidative stress, this study shows that melatonin may by acting by several potential mechanisms, mediated by receptor-dependent and receptor-independent pathways. One important property to consider is its scavenging capacity, neutralizing either ROS or RNS [82]. On the other hand, as we showed the MT-1 and MT-2 expression in cardiac tissue and vessels, the observed effects can be mediated via receptors. The activation of both receptors has been shown to modulate immediate early gene transcription of antioxidant enzymes [83,84,85]. However, the effects of simultaneous activation of both MT-1 and MT-2 are still under debate and may lead to additive, synergistic, or opposing responses that amplify or diminish each other [61].

Interestingly, previous studies have shown sex differences in fetal [86] and postnatal [87] cardiovascular responses to perinatal hypoxia, even in adulthood [88]. In the present study, the group composition was C:1Female/4Male; M:3Female/2Male. With these numbers we were unable to assess sex differences, which we considered as a limitation of our study. However, future studies should focus on assessing sex differences.

## 5. Conclusions

The night-time oral administration of melatonin in the first few postnatal weeks decreased the main pro-oxidant ROS sources at the cellular level, reducing oxidative stress and reinforcing the antioxidant status of the right ventricle in chronically hypoxic newborns with PAHN. Mechanisms underlying this protection by melatonin include the activation of antioxidant enzymes and modulation of pro-oxidant sources in RV. We consider that the capacity of an antenatal melatonin treatment to modulate postnatal cardiac oxidative balance is a remarkable finding, as oxidative stress is a hallmark of myocardial pathophysiology. In addition, the results are shown in a translational animal model that represents perinatal conditions in humans [35,36]. Therefore, our findings support melatonin as a plausible treatment in neonatal diseases that coexist with hypoxia and oxidative stress, with human translational potential to protect babies against cardiac disease induced by developmental chronic hypoxia.

## Figures and Tables

**Figure 1 antioxidants-10-01658-f001:**
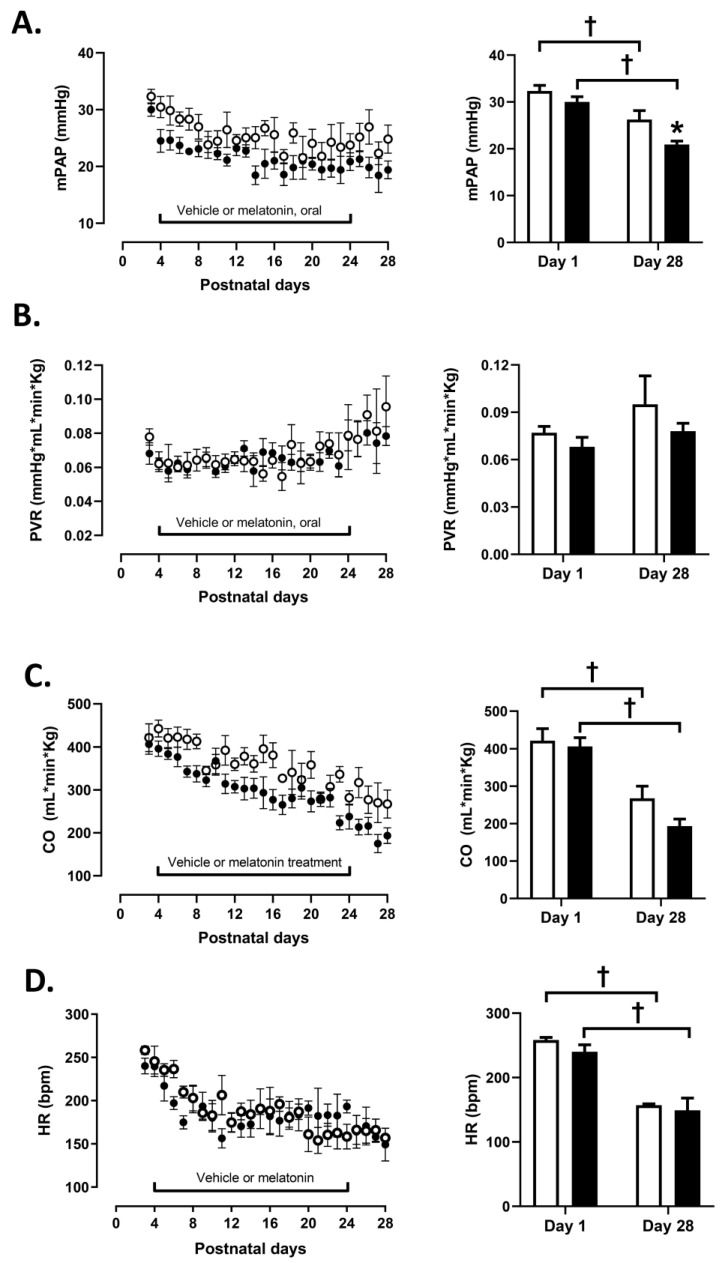
Cardiopulmonary variables during neonatal period. Postnatal cardiopulmonary evolution during the experimental period of the mean pulmonary arterial pressure (mPAP, (**A**), pulmonary vascular resistance (PVR, (**B**)), cardiac output (CO, (**C**)), and heart rate (HR, (**D**)) for controls (C, white symbols, bars) and melatonin treated (M, black symbols, bars). Histograms show the comparison between postnatal day 1 and 28. Data expressed as mean ± S.E.M. Significant differences (*p* ≤ 0.05): * vs. CN at equivalent days; † vs. day 1 of the same group.

**Figure 2 antioxidants-10-01658-f002:**
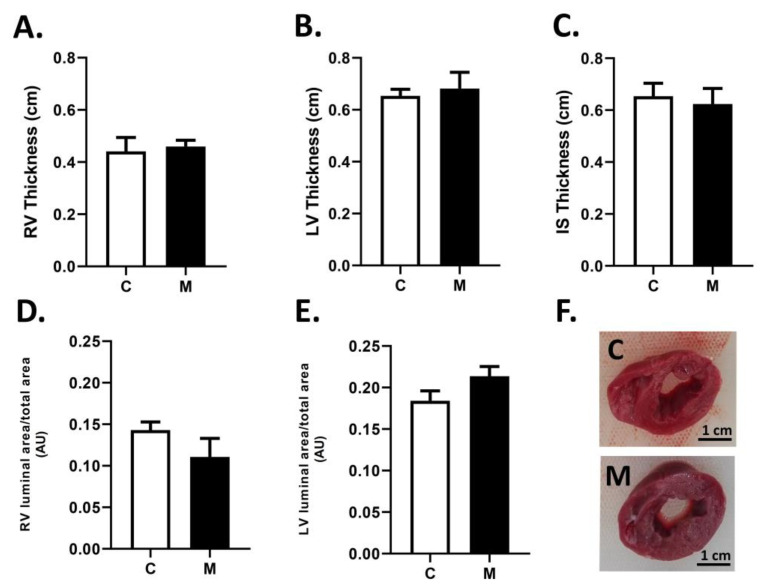
Cardiac biometry at the end of the experimental period. Thickness of right ventricle free wall (RV, (**A**)), left ventricle free wall (LV, (**B**)) and interventricular septum (IS, (**C**)); luminal:total area ratios for the right (**D**) and left (**E**) ventricles, at 29 days old. Representative images are shown (**F**). Bar in the photographs = 1 cm. Groups are control (C, white bars) and melatonin treated (M, black bars) lambs. Data expressed as mean ± S.E.M.

**Figure 3 antioxidants-10-01658-f003:**
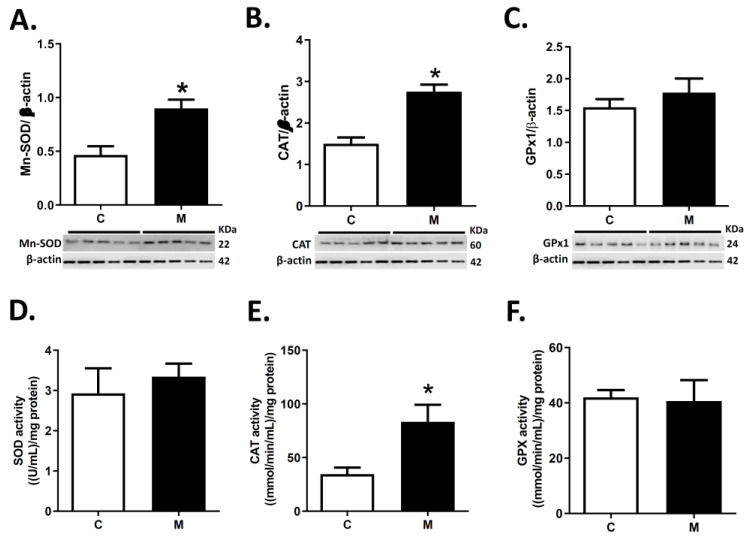
Enzymatic antioxidant expression and activity in right ventricle. Superoxide dismutase (SOD, (**A**)), Catalase (CAT, (**B**)) and Glutathione peroxidase (GPX, (**C**)) expression; and SOD (**D**), CAT (**E**) and GPX (**F**) activities, in right ventricle at 29 days old. Groups are control (C, white bars) and melatonin treated (M, black bars) lambs. Data expressed as mean ± S.E.M. Significant differences (*p* ≤ 0.05): * vs. CN.

**Figure 4 antioxidants-10-01658-f004:**
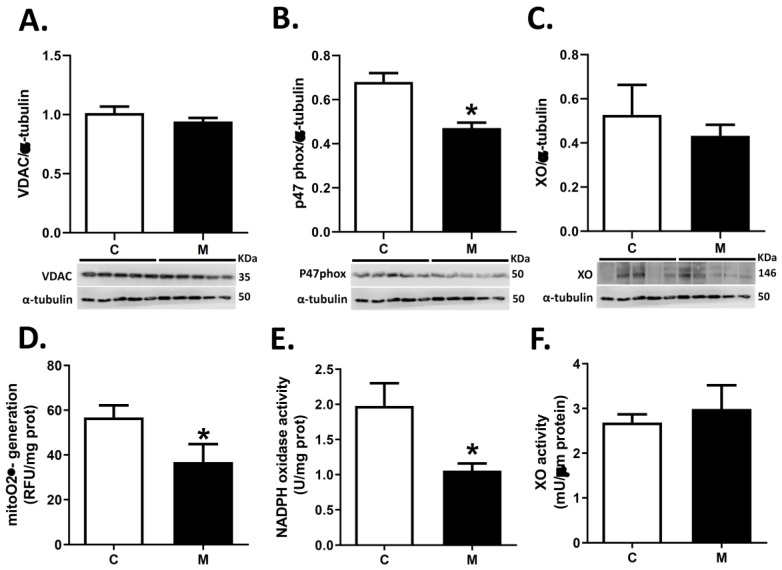
Cellular pro-oxidant sources in right ventricle. Reactive oxygen species estimated generation from mitochondrial (**A**,**D**), NADPH oxidase (**B**,**E**) and xanthine oxidase (**C**,**F**) activity and expression in right ventricle at 29 days old. Groups are control (C, white bars) and melatonin treated (M, black bars) lambs. Data expressed as mean ± S.E.M. Significant differences (*p* ≤ 0.05): * vs. CN.

**Figure 5 antioxidants-10-01658-f005:**
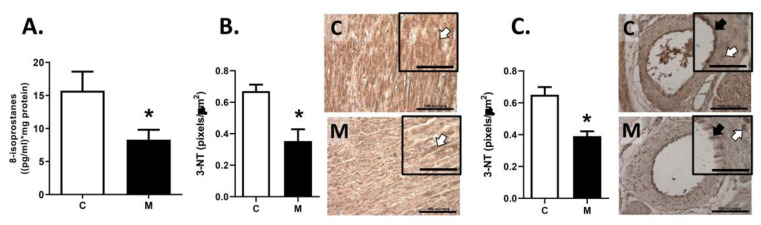
Oxidative stress in right ventricle. 8-isoprostanes (**A**) levels in right ventricle, 3-nitrotirosine (3-NT) immunostaining in RV free wall (**B**) and RV endocardial arteries (**C**) as oxidative markers in 29 days old lambs. Groups are control (C, white bars) and melatonin treated (M, black bars). Representative micrographs are shown for each analysis with bar = 100 μm. Zoom box inside the micrographs with bar = 50 μm; white and black arrows show immunopositivity in muscle and endothelial areas, respectively. Data expressed as mean ± S.E.M. Significant differences (*p* ≤ 0.05): * vs. CN.

**Figure 6 antioxidants-10-01658-f006:**
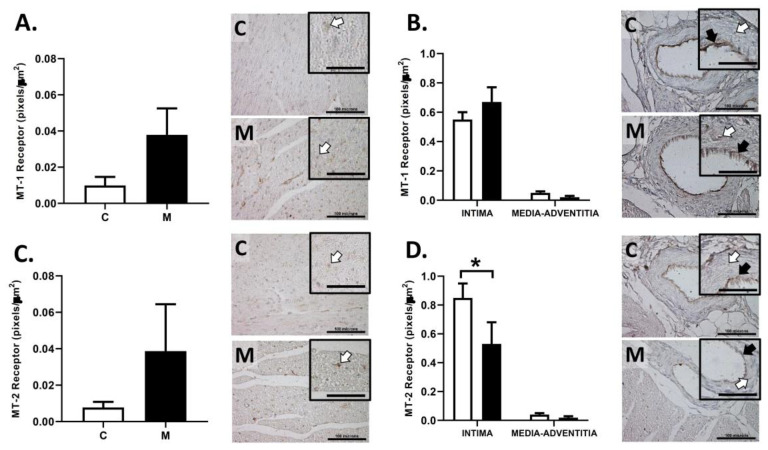
Melatonin receptors in right ventricle. Melatonin receptor immunostaining in right ventricle wall and endocardial arteries, for MT-1 (**A**,**B**) and MT-2 (**C**,**D**), in 29-day-old lambs. Groups are control (C, white bars) and melatonin treated (M, black bars). Representative micrographs are shown for each analysis with bar = 100 μm. Zoom box inside the micrograph with bar = 50 μm; white and black arrows show immunopositivity in muscle and endothelial areas, respectively. Data expressed as mean ± S.E.M. Significant differences (*p* ≤ 0.05): * vs. CN.

## Data Availability

Data is contained within the article.

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
