# Peer review of "Melatonin Reduces Oxidative Stress in the Right Ventricle of Newborn Sheep Gestated under Chronic Hypoxia"

_antioxidants, 2021, doi:10.3390/antiox10111658_

Round 1

Reviewer 1 Report

The Authors in the present original article evaluated the effects of postnatal melatonin administration on cardiac morphofunction modifications and oxidative stress in an ovine model of pulmonary arterial hypertension and observed that melatonin decreased right ventricle oxidative stress, but it did not affect cardiac structure. Interestingly, They observed also melatonin receptor involvement in the melatonin mechanism of action(s) against the adverse effects in newborns of hypobaric pregnancies.

Good article. I think that the present manuscript has an interesting topic and it could be a starting point for a better understanding of the pathophysiological process(es) involved in pulmonary arterial hypertension and, interestingly, it provided important evidence that melatonin treatment may protect the neonatal cardiac function and redox balance .

In my opinion, this manuscript has a clear message, the rationale for the choice of the experimental model as well as the technical approaches used are appropriate. The obtained results are fully described and discussed.

However, I have some mandatory comments:

- In all the text check that abbreviations and editing errors (such as at lines 39 and 101) and clearly define the groups C and M the first time that are cited.

Introduction

- Some references that justify the sentences reported are missing;

- the term “neurohormone” to define melatonin in my opinion is not fully correct (indoleamine sould be better);

- briefly amplify the melatonin receptors description, focusing on MT1 and MT2 functions;

Materials and Methods

- Line 106: specify the sex of the animals used. Sex differences may be observed?

- line 142: specify if the morphometrical analyses was done in blind and if the observers were two o three;

- lines 145-147: specify the primary antibodies dilutions used;

- specify the absorbance used for the Elisa kits.

Results

- Lines 228-232: describe the data in the order in which were presented at Fig. 3;

- Fig. 6: the immunohistochemical photomicrographs are a little bit dark and a zoom of the immunopositivity may be useful for the readers (especially to show endothelial positivity).

Discussion

- Briefly discuss the possible effects on left ventricule or add also in the Discussion the rationale due to it is not evaluated in the present study;

- discuss why melatonin treatment not affected cardiac structure,

- a simple schematic graph that summarized the data obtain should be useful for the readers.

Author Response

The Authors in the present original article evaluated the effects of postnatal melatonin administration on cardiac morphofunction modifications and oxidative stress in an ovine model of pulmonary arterial hypertension and observed that melatonin decreased right ventricle oxidative stress, but it did not affect cardiac structure. Interestingly, They observed also melatonin receptor involvement in the melatonin mechanism of action(s) against the adverse effects in newborns of hypobaric pregnancies.

Good article. I think that the present manuscript has an interesting topic and it could be a starting point for a better understanding of the pathophysiological process(es) involved in pulmonary arterial hypertension and, interestingly, it provided important evidence that melatonin treatment may protect the neonatal cardiac function and redox balance .

In my opinion, this manuscript has a clear message, the rationale for the choice of the experimental model as well as the technical approaches used are appropriate. The obtained results are fully described and discussed.

Comment: Thanks for the comment. We agree that melatonin effects might be therapeutically relevant, and we appreciate the reviewer constructive criticisms.

However, I have some mandatory comments:

- In all the text check that abbreviations and editing errors (such as at lines 39 and 101) and clearly define the groups C and M the first time that are cited.

Response: Thanks for the observation. We have double-checked the text, correcting errors and including abbreviations definitions.

Introduction

- Some references that justify the sentences reported are missing;

Response: We added references in some of the sentences.

- the term “neurohormone” to define melatonin in my opinion is not fully correct (indoleamine sould be better);

Response: Thanks for the comment. We agreed and replaced neurohormone by indoleamine, as suggested (Line 88).

- briefly amplify the melatonin receptors description, focusing on MT1 and MT2 functions;

Response: Thanks for the observation. We have added more effects of melatonin, highlighting MT1 and MT2 functions (Lines 97-107).

Materials and Methods

- Line 106: specify the sex of the animals used. Sex differences may be observed?

Response: This is an interesting point. The groups consisted in C=1F/4M; M: 3F/2M. With this numbers, we are unable to assess sex differences. However, we are currently applying for a project that will assess sex differences. The F/M ratio was added in the text (lines 128-130) and discussed (lines 471-475).

- line 142: specify if the morphometrical analyses was done in blind and if the observers were two o three;

Response: All analyses were performed by two independent observers, blind to the treatment groups. This sentence was added (Line 161).

- lines 145-147: specify the primary antibodies dilutions used;

Response: Done (section 2.4, lines 164-183)

- specify the absorbance used for the Elisa kits.

 Response: Done (sections 2.4, lines 164-177)

Results

- Lines 228-232: describe the data in the order in which were presented at Fig. 3;

 Response: Figure 3 legend was modified to be consistent with results description (Lines 292-296).

- Fig. 6: the immunohistochemical photomicrographs are a little bit dark and a zoom of the immunopositivity may be useful for the readers (especially to show endothelial positivity).

Response: We have fixed figures 5 and 6 to better show immunopositivity, with a zoom box inside each micrograph. The legend of the figures has been amended consistently. We hope that the reviewer likes these new versions.

 Discussion

- Briefly discuss the possible effects on left ventricule or add also in the Discussion the rationale due to it is not evaluated in the present study;

Response: At the beginning of the discussion, we added the rationale of assessing only the right ventricle (Lines 320-322).

- discuss why melatonin treatment not affected cardiac structure,

Response: We added a sentence describing that presumably the hypoxic insult induces an intense pulmonary vasoconstriction that melatonin is unable to revert. If this is true, oxygen supplementation in the clinical setting should improve the effects of melatonin (Lines 368-375).

- a simple schematic graph that summarized the data obtain should be useful for the readers.

Response: This was proposed by a graphical abstract. Please see the graphical abstract figure. If it’s the Reviewers choice, we can include the graphical abstract as Figure 7.

Author Response

MANUSCRIPT ACCEPTED WITH MAJOR REVISION

THE ENGLISH IS VERY GOOD WITH MINOR CORRECTIONS THE TOPIC IS INTERESTING
The authors study the influence of Melatonin in the right ventricle of neonatal lambs, dealing with a very important topic such as cardiac remodeling and right ventricle dysfunction following pulmonary arterial hypertension. However, when we look into the mechanism of melatonin and the conclusions reported by the authors there are some issues that should be clarified.

1) The increase in catalase activity seems to be related to the increase level of catalase. However, the same result is not obtained with Mn-SOD, that displays increase level but not increase activity. The authors should explain this result.

Response: Thanks for the comment. In the first version of the manuscript, we stated that a possible explanation is that total activity of SOD results from the sum of different isoforms of the enzyme, that can be modified in different ways in their function, expression and structure by either melatonin or oxidative stress. In addition to this explanation, we added the following: Particularly, the dichotomy between SOD expression and activity can be explained from an experimental point of view since the blot (expression) involves only the mitochondrial isoform, and the activity kit involve the three isoforms. In this sense, it can be inferred that the expression of all three isoforms is differentially regulated. The cytoplasmic SOD1 isoform is practically constitutive according to Minc et al (PMID: 9867871), while the mitochondrial SOD2 and extracellular SOD3 isoforms are regulated by different stimuli, in particular the redox balance (ie. NF-kB, AP-1, and AP-2) (PMID: 19477268) and inflammation (particularly SOD3, PMID: 12885586). On the other hand, melatonin via a receptor (MTI or MT2) can regulate the activation of these transcriptional factors such as AP-1 or NFkB (PMID: 26514204). Therefore, melatonin is acting on SOD functional regulation; however, further studies need to be implemented to assess the expression of SOD and GPx isoforms in neonatal hearts after melatonin treatment (lines 392-401).

2) In Line 295 the authors say that “postnatal treatment with melatonin protects the neonatal cardiac function and redox balance against the adverse effects of hypobaric pregnancies”. However, the only functional property of the heart shown by the authors is the heart rate. Author should take into account other parameters that are impaired due to pressure overload, such as ejection fraction and fractional shortening.

Response: Thanks for the comment. The reviewer refers to echocardiographic variables, a methodology that we did not perform in these animals. However, if we calculate the stroke volume with the HR and CO values (both functional, shown in the manuscript), we did not found any differences between groups (data not shown). Therefore, we would like to leave this sentence as it is. We hope that the reviewer agrees with us.

3) A proper measurement of ROS levels in the heart was not performed. Authors should show directly that heart slices of M-treated animals display less ROS levels (e.g., by staining, with DCF)

Response: Thanks for the comment. We performed two different approaches to determine oxidative and nitrosative stress in heart tissue. The first one was by determining 8-isoprostanes in tissue homogenate, supported by several studies as a reliable marker of oxidative stress
(PMID: 11350785, PMID: 11139135, PMID: 9420453). The second one is by determining 3-NT, to quantify nitrosative stress, that is closely related with oxidative stress (PMID: 24036104, PMID: 17049313). Interestingly, previous studies have shown that both markers are related to oxidative stress (PMID: 12539002; PMID: 17049313). Therefore, we think that using these 2 biomarkers is solid enough to demonstrate that melatonin is reducing cardiac oxidative stress.

4) The oxidation of NADPH by NADPH oxidase has been performed in the presence of Apocynin (Data not shown). However, Apocynin has been described to be not selective for NADPH oxidase and to act as an antioxidant per se (PMID: 18086956). Authors should repeat the experiment in the presence of a selective NADPH oxidase inhibitor, such as GSK2795039. I suggest this inhibitor given the experiment performed with p47phox that is a NOX2 subunit.

Response: Thanks for the observation. Unfortunately, we cannot repeat the experiments or perform new experiments, as our Labs are close following institutional instructions in response to COVID-19 pandemic. Therefore, we propose to include the following paragraph in the discussion to address the concern raised by the reviewer: "The NOX activity assay used apocynin as NOX inhibitor to create a negative control. Whilst apocynin has been widely used as NOX inhibitor in neurons, blood vessels and heart among other tissues [PMID: 30798073, 30771751, 27782741] it has been also questioned because of its antioxidant activity per se [PMID: 18086956]. We cannot completely rule out if apocynin acts as an antioxidant in addition to its action as NOX inhibitor in our assay; however, the similar 30 - 40 % decrease of both NOX activity and expression induced by melatonin in right ventricle suggest that our functional measurements are correct.".

Minor points:
1) Line 39: there is a missing parenthesis.
Response: Fixed (Line 39).
2) Line 95: there is a missing full stop.
Response: Fixed (Line 97).

Reviewer 3 Report

This is an excellent study on the reduced oxidative stress in the right ventricle of newborn sheep gestated under chronic hypoxia after melatonin administration that presents many new highly relevant findings on the mechanisms and mediators of the antioxidant effects of the indolamine.

Introduction

The introduction is fully sufficient and all the relevant previous work on the topic is considered. The relevance of their study is clearly outlined. The authors also explain why the findings of this study are novel and of relevance to their field of research. 

The authors could add a few sentencesin a brief paragraph on the validity of the animal model used by discussing the previous study in more detail. This would introduce this specific work better.

Materials and methods

Materials and methods are well described in detail so that the peers can reproduce the study. The applied methods are fully validated and the parameters measured cover the issues addressed. State of the art methods are used.

The experimental design is elegant and efficient. The protocols used allow for an indept analysis of the measured parameters. The analysis allow for a decisive evaluation of the experimental changes in the groups.

Results

The results are well presented and documented. Their description is precise and correct.

Discussion

As stated in my review, the reasons on why melatonin failed to alter cardiac structure in this specific study are not suffiently discussed and explained. The manuscript could be improved by a paragraph that addresses the findings of this specific study.

This can be done in comparison to the previous one and should also touch the issue of why this topic is relevant (as correctly mentioned in the introduction and also, but not suffiently addressed in their current discussion.

Otherwise, the discussion is concise and sufficient. The authors mention the similiarities and differences of their findings in the context of the findings of others. This is an excellent discussion.

They conclusively discuss their data and findings. The discussion is broad and helpful in understanding the relevance of the findings presented in their study.

Conclusions

The conclusions are backed by the results. Their findings are of utmost interdisciplinary importance and relevance. The authors may add a final sentence demonstrating the relevance of their work for translational biomedicine.

Here again, they may stress the validity of their approach, i.e. the relevance of their specific animal model for human cardiac disease and the prevention of and protection against developmental chronic hypoxia in babies.

General comment

Could you please discuss in more detail, why melatonin administration did not affect cardiac structure in this study?

Specific comments with suggestions for very minor changes

Line 96: Period at the end of the sentence is missing. Please add.

Line 220: ... than in the control group ...

Line 229 ff: Abbreviations C and M are not introduced.

Author Response

This is an excellent study on the reduced oxidative stress in the right ventricle of newborn sheep gestated under chronic hypoxia after melatonin administration that presents many new highly relevant findings on the mechanisms and mediators of the antioxidant effects of the indolamine.

Comment: Thanks for the comment. We appreciate that you found our study interesting.

Introduction

The introduction is fully sufficient and all the relevant previous work on the topic is considered. The relevance of their study is clearly outlined. The authors also explain why the findings of this study are novel and of relevance to their field of research. 

The authors could add a few sentences in a brief paragraph on the validity of the animal model used by discussing the previous study in more detail. This would introduce this specific work better.

Response: Thanks for the comment. We added a couple of sentences describing the usefulness of the lamb model (Lines 112-117).

Materials and methods

Materials and methods are well described in detail so that the peers can reproduce the study. The applied methods are fully validated and the parameters measured cover the issues addressed. State of the art methods are used.

The experimental design is elegant and efficient. The protocols used allow for an indept analysis of the measured parameters. The analysis allow for a decisive evaluation of the experimental changes in the groups.

Results

The results are well presented and documented. Their description is precise and correct.

Discussion

As stated in my review, the reasons on why melatonin failed to alter cardiac structure in this specific study are not suffiently discussed and explained. The manuscript could be improved by a paragraph that addresses the findings of this specific study.

This can be done in comparison to the previous one and should also touch the issue of why this topic is relevant (as correctly mentioned in the introduction and also, but not suffiently addressed in their current discussion.

Otherwise, the discussion is concise and sufficient. The authors mention the similiarities and differences of their findings in the context of the findings of others. This is an excellent discussion.

They conclusively discuss their data and findings. The discussion is broad and helpful in understanding the relevance of the findings presented in their study.

Response: In the discussion, we expand a possible explanation of the lack of melatonin effects on cardiac structure (Lines 368-375).

Conclusions

The conclusions are backed by the results. Their findings are of utmost interdisciplinary importance and relevance. The authors may add a final sentence demonstrating the relevance of their work for translational biomedicine.

Here again, they may stress the validity of their approach, i.e. the relevance of their specific animal model for human cardiac disease and the prevention of and protection against developmental chronic hypoxia in babies.

Response: Thanks for the comment. We added a sentence to highlight the relevance of the study for translational medicine (Lines 484-485).

General comment

Could you please discuss in more detail, why melatonin administration did not affect cardiac structure in this study?

 Response: We added a possible explanation of the lack of melatonin effects on cardiac structure (Lines 368-375).

Specific comments with suggestions for very minor changes

Line 96: Period at the end of the sentence is missing. Please add.

Response: Done (Line  97).

Line 220: ... than in the control group ...

Response: Done (Line 246).

Line 229 ff: Abbreviations C and M are not introduced.

Response: C & M definitions are included in Material and Methods section (Line 128-130).

Round 2

Reviewer 2 Report

I thanks the author for their answers to my questions. I understand that Covid pandemic has created several problems. I firmly think that points 3 and 4 should be covered, maybe by following experimeriments that could be performed for the next work by authors.

Overall, I accept the answers and I think that the manuscript can be accepted in present form.